# Anaesthetic Techniques and Strategies: Do They Influence Oncological Outcomes?

Liam Murphy [1,*], John Shaker [1] and Donal J. Buggy [1,2,3]

1   Department of Anaesthesiology & Perioperative Medicine, Mater University Hospital, School of Medicine, University College Dublin, D07 R2WY Dublin, Ireland; johnshaker@mater.ie (J.S.); donal.buggy@ucd.ie (D.J.B.)
2   European Society of Anaesthesiology and Intensive Care Onco-Anaesthesiology Research Group, 24 Rue des Comédiens, B-1000 Brussels, Belgium
3   Outcomes Research, Cleveland Clinic, Cleveland, OH 44195, USA
*   Correspondence: liammurphy44@hotmail.com

**Abstract:** Background: With the global disease burden of cancer increasing, and with at least 60% of cancer patients requiring surgery and, hence, anaesthesia over their disease course, the question of whether anaesthetic and analgesia techniques during primary cancer resection surgery might influence long term oncological outcomes assumes high priority. Methods: We searched the available literature linking anaesthetic-analgesic techniques and strategies during tumour resection surgery to oncological outcomes and synthesised this narrative review, predominantly using studies published since 2019. Current evidence is presented around opioids, regional anaesthesia, propofol total intravenous anaesthesia (TIVA) and volatile anaesthesia, dexamethasone, dexmedetomidine, non-steroidal anti-inflammatory medications and beta-blockers. Conclusions: The research base in onco-anaesthesia is expanding. There continue to be few sufficiently powered RCTs, which are necessary to confirm a causal link between any perioperative intervention and long-term oncologic outcome. In the absence of any convincing Level 1 recommending a change in practice, long-term oncologic benefit should not be part of the decision on choice of anaesthetic technique for tumour resection surgery.

**Keywords:** cancer recurrence; cancer metastasis; cancer surgery; anaesthesia; onco-anaesthesia; surgery; postoperative analgesia; regional anaesthesia; total intravenous anaesthesia; volatile anaesthesia

## 1. Introduction

The global disease burden of cancer is significant; it is responsible for 10 million deaths globally, which is an increase of 21% since 2010 [1]. This trend is projected to continue until at least 2040 and is the result of globally aging populations [2]. It is well recognised that deaths from cancer do not fully illustrate the impact of cancer on patients and their families and on global health services.

Management of solid tumours can take the form of medical or surgical treatment, or a combination. As many as 60% of solid tumours are amenable to primary resection surgery with curative intent. Over 80% of patients with a cancer diagnosis will receive anaesthesia and surgery at some point in their disease journey, including diagnostic or palliative procedures.

Surgical intervention itself may play a role in oncological outcomes. While removal of the primary tumour is the mainstay of treatment for many solid cancers, inadvertent displacement of minimal residual disease, such as microscopic tumour cells, into the circulation during resection could potentially facilitate development of metastases [3]. In addition, the surgical stress response, which is characterised by modulation of the immune, inflammatory and adrenergic systems, may influence rates of cancer metastases and disease progression [4]. Complex interactions between multiple immune factors and

metastatic deposits play a role in the propagation of metastatic disease. These interactions present potential therapeutic avenues to minimise the risk of metastatic disease becoming established at the time of cancer surgery. This poses interesting questions regarding the impact of surgery on cancers and potential avenues for pharmacological intervention [5] (Figure 1).

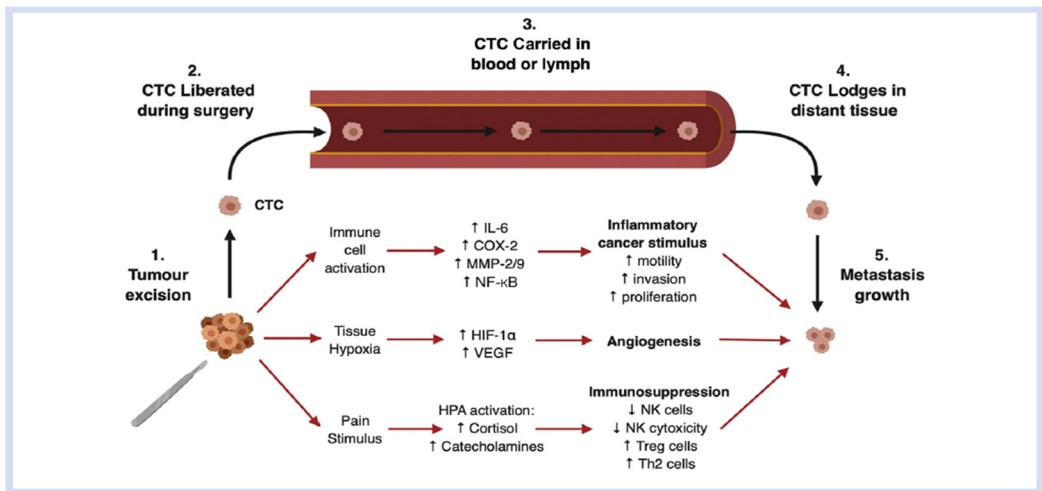

**Figure 1.** Schematic representation of the pathophysiological mechanisms induced by surgery that promote survival and growth of metastatic deposits formed by circulating tumour cells (CTCs) released intraoperatively. COX-2, cyclo-oxygenase-2; HIF, hypoxia-inducible factor; HPA, hypothalamic-pituitary-adrenal axis; IL-6, interleukin 6; MMP, matrix metalloprotease; NF-kB, nuclear factor kappa B; NK, natural killer cell; Th2, Type 2 helper T cell; Treg, regulatory T-cell; VEGF, vascular endothelial growth factor.

Anaesthetic techniques and strategies have also been implicated in recent years as both potentially harmful techniques that worsen oncological outcomes and potential therapeutic avenues that can minimise the risk of cancer recurrence and metastases [6]. Techniques examined include, but are not limited to, intra-operative opioid use, volatile inhalational anaesthesia and propofol total intravenous anaesthesia [TIVA], regional anaesthesia, the systemic use of amide local anaesthetics, dexamethasone and use of dexmedetomidine (a highly selective alpha-2 adrenergic agonist). To facilitate further research, a workgroup has determined Standardised End-Points (StEPs) to standardise measured primary and secondary outcomes in clinical trials of the effect of peri-operative interventions on oncologic outcomes [7]. Here, we review current evidence around a number of anaesthetic–analgesic techniques in relation to long-term cancer outcomes.

## 2. Opioids

The use of opioids is a mainstay of anaesthesia and perioperative analgesia in patients undergoing cancer surgery. Of primary interest are the immune-modulating effects of opioid medications, and how they may be of relevance in cancer immunology. The direct and indirect effects of opioids on cancer cells and the effects on cells involved in anti-tumour immunity, such as NK cells, macrophages and T-cells, are well described in a paper from Boland and Pockley [8]. Theoretically, opioid receptor expression on tumour cells can be implicated in cancer cell proliferation and cancer migration; therefore, the opioids we administer therapeutically after surgery could interact with tumour opioid receptors and increase tumour activity [9]. As described in a retrospective study [10], in metastatic prostate cancer, increased tumour MOR expression resulted in reduced progression-free and overall survival.

As a result, there has been a new focus on detailed genomic analyses of patients' excised tumour tissue and how individual patient tumour gene expression may interact

with perioperative opioid use during tumour resection surgery and subsequent oncological outcomes. This focus is welcomed because any true effects of anaesthetic-analgesic interventions during cancer surgery may be measurable only in defined tumour subtypes and on individual patients' tumour genomic expression [11].

However, in a retrospective study of over 8000 patient tumours samples [12], utilising the Cancer Genome Atlas, no correlation between oncological recurrence and opioid receptor expression on a wide variety of tumour cells was shown. Additionally, a further retrospective cohort study [13], examining patients with stage I–III colorectal cancer demonstrated that despite increased expression of Opioid Growth Factor Receptor and mu-opioid receptor (MOR), this did not translate into an association with altered recurrence rates. Furthermore, a randomised control trial ($n = 146$) studying patients undergoing radical prostatectomy for intermediate and high D'Amico risk prostate cancer, were randomised to receive either opioid-free or opioid-based anaesthesia. No statistically different biochemical recurrence rates or recurrence-free survival was observed, but this trial was underpowered to detect the latter [14].

A different approach has been taken by a New York group, who evaluated the influence of patient-specific tumour gene expression on responses to intraoperative interventions during tumour resection. A retrospective study focusing on intraoperative opioid use during primary resection of stage I–III colon adenocarcinoma found that tumour recurrence was lower in patients with higher cumulative intra-operative opioid dose. In addition to this, immunohistochemistry analysis identified that, in tumours with diminished DNA mismatch repair (MMR) ability, there was a stark reduction in recurrence compared with tumours with preserved DNA MMR [15]. A separate retrospective analysis [16] studied the same principle, evaluating triple negative breast tumours from 1143 patients who had undergone surgical resection; the results showed that the genetic make-up of their individual tumours expressed some tumour opioid receptors. This analysis illustrated not only downregulation, or even absence, of pro-tumour receptors in the presence of opioid agonism, but also an association with the upregulation of anti-tumour receptors. Taken together, these findings suggest an association between a protective effect of intraoperative opioids and recurrence-free survival in triple-negative breast cancer, but not with improved overall survival.

Another retrospective study of 239 patients undergoing resection of hepatocellular carcinoma [17] examined the effect of low-dose versus high-dose post-operative morphine needs on oncologic outcomes. High dose (86 mg morphine equivalent) was deemed above the median value of opioid use across both arms of the study. Patients receiving the high-dose morphine had an increased all-cause mortality, but this did not correlate with cancer recurrence risk. Beyond its retrospective design, this study had further limitations, including small sample size, lack of tumour genomic testing and not accounting for complexity of surgery, which could explain the increased intra- and post-operative opioid requirements. A separate retrospective study which utilised tumour genomic sequencing [18] examined 740 patients with stage I–III lung adenocarcinoma. This demonstrated a varied impact of higher intra-operative opioid use depending on tumour gene expression.

Despite the widespread use of opioids during surgical resection of solid tumours, the level of understanding of how these medications influence oncological outcomes remains suboptimal. While laboratory research conducted a decade ago initially suggested that opioids might have a detrimental impact on cancer, facilitating tumour cell survival, new research examining tumour sub-types and intra-tumoral gene expression highlight how nuanced the potential effect of perioperative opioids during cancer resection surgery on oncological outcomes may be.

## 3. Regional Anaesthesia Techniques

Regional anaesthesia techniques, both central neuraxial and peripheral nerve blocks, have been associated with improved oncological outcomes in some observational studies. The basis of this hypothesis is that regional anaesthesia preserves immune function and

reduces surgical stress peri-operatively, reducing postoperative inflammation and thus reducing the risk of cancer recurrence by inhibiting pro-tumour pathways [19]. Potential mechanisms for modulation of this pro-tumour pathway are examined in depth in a recent paper by Li et al., who described the effects that local anaesthetics may have on tumour cells directly, catecholamine release, voltage gated sodium channels, systemic angiogenic factor concentrations and a reduction in postoperative pain and opioid use, as well as how factors may influence oncological outcomes [20]. Furthermore, regional anaesthesia techniques allow for potential reduction in exposure to volatile anaesthetic agents, which some translational research suggests may be of benefit in reducing cancer recurrence.

A retrospective study from Danish national databases, which examined 11,618 patients with colorectal cancer who had surgery between 2004 and 2018 and were followed over a median duration of 58 months, found that epidural anaesthesia was not quite associated with lowered cancer recurrence rates compared with patients receiving GA alone (hazard ratio, 0.91; 95% CI, 0.82 to 1.02) [21]. A smaller retrospective study on 218 patients with pancreatic cancer who underwent resection with curative intent demonstrated no alteration to overall survival or recurrence rate when epidural anaesthesia was utilised (HR: 0.98; 95% CI, 0.78–1.24%; $p = 0.87$ and HR: 1.02; 95% CI, 0.82–1.27%; $p = 0.85$), respectively [22]. This data should be interpreted in the context of this study being underpowered.

However, after over a decade of conflicting findings from observational studies, a large, multi-centre randomised control trial evaluating the effect of paravertebral regional anaesthesia–analgesia versus volatile anaesthesia with opioid analgesia on oncologic outcomes was conducted in women undergoing primary breast cancer resection. Some 2108 patients with breast cancer were randomised to either paravertebral regional with propofol general anaesthesia or volatile general anaesthesia with opioid analgesia and followed up for a median of over 3 years The incidence of breast cancer recurrence was approximately 10% in both groups, indicating robust neutral findings [23].

A modest trial of 180 patients with colorectal cancer undergoing primary resection was performed [24], in which patients were randomised to either general anaesthesia plus opioid-based patient-controlled anaesthesia or general anaesthesia plus thoracic epidural anaesthesia. The primary outcome was the surrogate end-point of return to intended oncologic treatment (RIOT). They reported no difference in RIOT, which is often a confounding factor in these studies, because a delay in RIOT can worsen cancer prognosis. There were also neutral findings regarding the effect of these analgesic techniques on cancer recurrence, although this study was underpowered to evaluate this. Interestingly, this study utilised an epidural regime which included an opioid with local anaesthetic, thus adding another confounding factor to their study. An alternative may have been to use a pure local anaesthetic epidural infusion to eliminate opioid effect on the tumour.

A further RCT that examined 40 patients with advanced ovarian cancer, comparing intraperitoneal ropivacaine and 0.9% saline with the primary outcome of RIOT, demonstrated that the intraperitoneal ropivacaine group achieved the primary outcome significantly sooner (median 21 (inter-quartile range 21–29) vs. 29 (inter-quartile range 21–40) days; $p = 0.021$) [25]. Whether this surrogate outcome measure (RIOT) translates into a measurable benefit in patient-centric oncologic outcomes remains to be seen in a properly powered RCT.

Another RCT with delirium as its primary end-point ($n = 1712$) undertook long-term follow up (median 66 months) of its patients who had a variety of non-cardiothoracic and abdominal cancers and who had been randomised to either epidural and general anaesthesia or general anaesthesia alone for surgical resection [26]. Epidural anaesthesia reduced the 7-day incidence of delirium. However, there was no statistically significant difference between the groups in terms of overall survival (HR 1.07; 95% CI, 0.92 to 1.24; $p = 0.408$), cancer-specific long-term survival (HR 1.09; 95% CI, 0.93 to 1.28; $p = 0.290$) or recurrence-free survival (HR 0.97; 95% CI, 0.84 to 1.12; $p = 0.692$). This study has significant limitations, however, with 8% of included participants having non-cancer surgery, albeit evenly distributed between both groups. Secondly, the epidural anaesthesia group also

received sufentanil in their epidural infusions, resulting in one group receiving a long-acting opioid (general anaesthesia alone) and one group receiving a short-acting opioid (epidural anaesthesia group). This study also was not originally designed to examine long-term survival and is therefore underpowered to elicit subtle differences in cancer survival. Nonetheless, it signals no meaningful effect of epidural anaesthesia on long term oncologic outcomes.

A further randomised control trial examining 400 patients undergoing a Video-assisted Thoracoscopic Surgery (VATS) for lung cancer randomised patients to general anaesthesia with or without epidural anaesthesia over a median follow up of 32 months and found no significant difference in overall survival (HR 1.12; 95% CI, 0.64 to 1.96; $p = 0.697$) or recurrence-free survival (HR 0.90; 95% CI, 0.60 to 1.35; $p = 0.608$) between the groups [27]. The epidural infusion in this study also contained sufentanil.

Aggregating all these findings, it can be definitively concluded that, while regional anaesthesia undoubtedly has many benefits for patients undergoing cancer surgery, the evidence base now clearly indicates that it has a neutral influence on oncologic outcomes.

## 4. Propofol Total Intravenous Anaesthesia (TIVA) and Volatile Anaesthesia

Laboratory studies had indicated a signal that the effect of propofol on tumour cell biology, inflammation and immune function might be more favourable in preventing recurrence with propofol compared with volatile agents [28]. This has been supported by a number of observational clinical reports. Initially, a retrospective study including 7000 patients with various cancer diagnoses after propensity matching suggested an association between clinically significant improvement in survival with propofol TIVA, in comparison to inhalational anaesthesia with multivariate analysis demonstrating higher risk of death in the inhalational group (HR 1.46, 95% CI, 1.29 to 1.66) [29]. Meta-analyses since this initial retrospective study have supported this hypothesis, including 19 retrospective studies, which showed an association between propofol TIVA and improved disease-free survival versus inhalation anaesthesia [30]. This meta-analysis is compromised by its comprising multiple small retrospective studies.

A small randomised controlled trial that examined $n = 210$ patients demonstrated no statistically significant difference between propofol TIVA and volatile anaesthesia cohorts on postoperative circulating tumour cell counts (RR 1.27 [95% CI, 0.95 to 1.71]; $p = 0.103$) [31]. A smaller RCT ($n = 153$) studying colorectal adenocarcinoma and the impact of anaesthesia technique on circulating levels of Natural Killer immune cells and T-cells post-operatively found no difference between propofol TIVA and sevoflurane cohorts at 24 h (RR $-2.6$ [95% CI, $-6.2$ to 1.0]; $p = 0.151$) [32]. A small RCT examining peri-operative levels of markers NETosis (Neutrophil Extracellular Trapping (NETosis), a biomarker implicated in cancer progression and metastasis) in 40 patients with breast cancer demonstrated no difference between patients who had received regional anaesthesia or opioid analgesia during breast cancer resection [33].

As always, no number of observational studies can provide Level I evidence for a causal relationship between any anaesthetic technique and cancer recurrence. A number of RCTs are currently being conducted studying this area which will hopefully bring some clarity. Notably, the CAN study (Cancer and Anaesthesia), which focuses on breast cancer patients, is randomising breast cancer patients to propofol TIVA or sevoflurane and measuring long term cancer outcomes. Interim analysis from this study demonstrates no noted difference in overall survival; however, five-year follow up is not yet complete [34]. Recruitment is also currently ongoing for the VAPOR-C trial (Volatile Anaesthesia and Perioperative Outcomes Related to Cancer), which aims to recruit 3500 patients undergoing surgery for colorectal cancer or non-small cell lung cancer across multiple centres. This trial has disease-free survival as its primary outcome and is a $2 \times 2$ factorial design comparing both propofol TIVA with volatile anaesthesia and systemic lidocaine or placebo within each GA arm of the trial.

In summary, while a number of inherently limited retrospective studies have suggested benefits from propofol TIVA in overall survival, there is no data from any large RCT to date, which is necessary before any change in practice can be recommended.

## 5. Dexamethasone

Dexamethasone is a glucocorticosteroid often utilised during anaesthesia in the prevention of post operative nausea and vomiting. The potential impact this practice may have on oncological outcomes has been raised, yet clarity remains elusive. It is hypothesised that the immunosuppressive effects of a steroid could result in an increased likelihood of distant metastases. This complex signalling pathway remains poorly categorised however dexamethasone has been shown in a 2016 study to impact on immune cells with a lymphodepletive effect noted, primarily effecting CD4$^+$ T cells but also CD8$^+$ T cells, dendritic cells and regulatory T cells (Tregs) [35]. Alternatively, potential benefits of dexamethasone's anti-inflammatory and anti-angiogenesis properties may in fact inhibit cancer and include improved oncological outcomes with increased metastasis free survival.

This uncertain picture is illustrated in a study utilising xenograft mouse models [36] that examined the effect of glucocorticoids on breast cancer progression. They described the complex signalling pathway which dexamethasone has in different tumour cells and how this makes interpretation of oncological effects difficult. This study concluded that low-dose dexamethasone may have beneficial effects reducing tumour growth and mitigating risk of metastases, while high-dose dexamethasone may in fact cause harm, increasing the risk of breast cancer progression.

A retrospective, cohort study of 2628 patients who underwent breast cancer surgery found that the 8.5% of patients who received single dose dexamethasone had no change in risk of recurrence (HR 1.389; 95% CI, 0.904–2.132; $p = 0.133$) or mortality (HR 1.506; 95% CI, 0.886–2.561; $p = 0.130$) on propensity scoring [37]. A separate study of 373 patients with pancreatic ductal adenocarcinoma elicited similar results, concluding that there was no improvement in recurrence-free (17 vs. 17 months; $p = 0.99$) or overall (46 vs. 43 months; $p = 0.90$) survival amongst the 60% of patients who received dexamethasone [38].

Interestingly, a retrospective study of 185 patients with bladder cancer who underwent radical resection concluded that patients who received glucocorticoids had a shortened metastasis-free survival time (HR 1.790; $p = 0.030$) when the compound variable of intra-operative blood transfusion was excluded from the analysis [39].

In contrast, a recent, large retrospective study involving >30,000 patients who had a solid cancer resection found that, in cancer patients not amenable to immune modulator therapy, peri-operative dexamethasone was associated with decreased one-year mortality (HR 0.82; 95% CI, 0.69–0.96; $p = 0.016$) and cancer recurrence (Adjusted Odds Ratio 1.28; 95% CI, 1.18–1.39; $p < 0.001$) [40]. However, this does not prove a causal link, which requires an RCT.

Therefore, while dexamethasone and its effects on oncological outcomes continue to be researched, there is currently little evidence justifying change in clinical anaesthesia practice on the basis of a benefit in cancer outcomes.

## 6. Dexmedetomidine

Dexmedetomidine is a highly selective alpha 2 adrenergic agonist which initially was licensed for sedation in intensive care units in 1999 and has since become more included in the realm of anaesthesia. Even at that time, it was known that dexmedetomidine preserved Natural Killer cell function peri-operatively, likely due to cortisol level suppression [41]. This preservation of NK cells was hypothesised as a mechanism for improving oncological outcomes during surgery, preventing cancer progression.

While dexmedetomidine theoretically makes sense as an adjunct during onco-anaesthesia due to its NK cell preservation and sympatholytic and anti-inflammatory properties, the evidence base does not support its adoption into clinical practice for oncological purposes. Additionally, the complex interactions between the immune system and tumour growth and

metastases should be considered when considering any immune-modulating medication used during the high-risk period, from a metastatic point of view, that is cancer surgery.

A laboratory investigation utilising ovarian cancer xenograft mouse models, found that NK cell function recovered faster in the dexmedetomidine group and lowered tumour burden at four weeks [42]. However, an RCT involving 100 patients with uterine cancer demonstrated no favourable impacts on NK cells ($p = 0.496$) and no statistically significant difference in rates of recurrence ($p = 0.227$) or death within two years ($p = 0.318$) [43]. Given the small sample size, this study was underpowered to elicit subtle differences in recurrence. However, the rates of both end points were lower in the dexmedetomidine cohort; (16.3% vs. 8.7%) and (6.7% vs. 2.2%), respectively.

Most recently, though, a follow-up analysis of an RCT on $n = 620$ older cancer surgical patients originally designed with a non-cancer primary end-point, found a benefit of dexmedetomidine infusion during anaesthesia on recurrence-free survival and event-free survival [44]. Median follow-up time was 42 months. While overall survival did not differ, there were 49/309 (16%) deaths with dexmedetomidine versus 63/310 (20%) with placebo (adjusted hazard ratio [HR] 0.78, 95% CI 0.53–1.13, $p = 0.187$). Recurrence-free survival was also apparently improved with dexmedetomidine (68/309 (22%) events with dexmedetomidine versus 98/310 [32%] with placebo; adjusted HR 0.67, 95% CI 0.49–0.92, $p = 0.012$). Event-free survival was also improved with dexmedetomidine (120/309 (39%) events with dexmedetomidine versus 145/310 [47%] with placebo; adjusted HR 0.78, 95% CI 0.61–1.00). While this is encouraging, confirmation of this finding in another RCT where oncologic outcome is the primary end-point is warranted.

### 7. NSAIDs/COX 2 Inhibitors and Beta Blockers

NSAIDs and their potential impact on oncological outcomes have been extensively researched in laboratory studies and observational, retrospective studies. However, there remains a relative scarcity of well-powered prospective randomised control trials to justify adjusting anaesthesia practice regarding NSAIDs to improve oncological outcomes [45]. The anti-inflammatory properties exhibited by these drugs are suggested to reduce cancer cell resistance to common treatment modalities, such as chemo and radio-therapy, by inhibiting the cyclo-oxygenase 2 receptor, which is often over expressed on cancer cells. The expression of cyclo-oxygenase 2 receptors on cancer cells is shown to promote carcinogenesis mediated through its effects on cancer stem cell-like activity, apoptotic resistance, proliferation, angiogenesis, inflammation, invasion and metastasis of cancer cells [46].

Two relatively sizable studies that examined extended courses of NSAIDs after initial surgical management have failed to prove any benefit of protracted NSAIDs exposure. Firstly, a RCT including >2500 patients with stage 3 colorectal cancer were randomised to receive either Celecoxib 400 mg once daily or placebo for 3 years in conjunction with FOLFOX adjuvant chemotherapy [47]. This study elicited no difference in three-year disease-free survival (HR for disease recurrence or death, 0.89; 95% CI, 0.76–1.03; $p = 0.12$) or in five-year overall survival (HR for death, 0.86; 95% CI, 0.72–1.04; $p = 0.13$). A second RCT that examined >2600 patients with ERBB2 negative breast cancer also demonstrated no benefit in five-year disease-free survival (unadjusted HR 0.97; 95% CI, 0.80–1.17; log-rank $p = 0.75$) with a treatment course of Celecoxib 400 mg once daily for a period of two years [48].

Recent RCTs have been attempting to add to the existing evidence base regarding the use of NSAIDs to improve oncological outcomes; however, these studies are difficult to interpret due to their small sample sizes. The first is a 34 patient study which randomised patients with colorectal cancer to receive either propranolol and etodolac (a CoX-2 inhibitor) for 20 days perioperatively and beginning 5 days prior to surgery, or placebo [49]. This study demonstrated a weakly favourable impact on tumour molecular markers of metastatic potential and also with a reduced rate of recurrence ($p = 0.05$) in the treatment cohort. A second RCT following 80 patients who underwent hepatectomy for hepato-cellular carcinoma classified patients into a treatment group of parecoxibsodium 40 mg and a

control group of placebo. This suggested that disease-free survival was significantly longer in the treatment arm (19.0 months, 95% confidence interval [CI], 9.8–28.2 vs. 14.0 months, 95% CI, 8.1–19.9; $p < 0.05$). Nonetheless, this did not translate to significantly increased overall survival time (36.0 months, 95% CI, 13.4–58.9 vs. 14.0 months, 95% CI, 10.6–25.4; $p > 0.05$) [50].

While recent additions of randomised control trials to the evidence base surrounding use of peri-operative NSAIDs and beta-blockers and their potential oncological benefits associated are welcome, and appear somewhat promising, enthusiasm is significantly tempered by the modest size of both trials. This serves to highlight a potentially promising therapeutic avenue that should be further explored with sufficiently powered trials.

Encouragingly, an RCT from a number of Indian centres among women undergoing breast cancer surgery with curative intent has just been reported. This group randomised almost 1600 women to an active arm (who received infiltration of amide local anaesthetic lidocaine 0.5 mg.kg, up to 4.5 mg.kg body weight, 7–10 min prior to surgical excision "LA") and compared them to a control group that did not receive this lidocaine infiltration ("No LA"). Median follow-up time was >5.5 years (68 months). In LA and no LA arms, 5-year DFS rates were 87% and 83% (hazard ratio [HR], 0.74; 95% CI, 0.58 to 0.95; $p = 0.017$) and 5-year OS rates were 90% and 86%, respectively (HR, 0.71; 95% CI, 0.53 to 0.94; $p = 0.019$). The impact of LA was similar in subgroups defined by menopausal status, tumour size, nodal metastases, and hormone receptor and human epidermal growth factor receptor 2 status. No adverse effects from lidocaine were observed [51]. This is the first trial to report a positive difference of a single perioperative intervention on long-term oncologic outcomes and will encourage ongoing efforts among anaesthesiologists and other clinicians to complete other trials, testing the long-term oncologic effects of various perioperative interventions during primary cancer surgery, in the field of onco-anaesthesiology (Table 1).

**Table 1.** Registered Randomised Controlled Trials evaluating anaesthetic-analgesic techniques and oncologic outcomes. DFS = Disease Free Survival; RFS = Recurrence Free Survival; OS = Overall Survival.

| Trial Number (Acronym) | Area of Focus | Type of Surgery | Number of Patients | Primary End-Point | Secondary End-Points | Est Year of Completion |
|---|---|---|---|---|---|---|
| NCT01975064 (CAN study) [32] | Propofol vs. Sevoflurane anaesthesia | Breast, Colon, Rectal Cancer surgery | 1700 | Overall Survival (OS) | OS 1 year | 2023 |
| NCT04316013 Volatile anaesthesia and perioperative outcomes related to cancer: The VAPOR-C Trial | 2 × 2 factorial design volatile vs. propofol TIVA; With and without IV lidocaine | Colorectal, non-small cell lung | 3500 | DFS | OS DAH-30 Postop complications | 2028 |
| NCT04449289 | Influence of intravenous lidocaine and peridural ropivacaine | Pancreatic surgery | 100 | 1- and 3-years recurrence rate after surgery | 1- and 3-years survival after surgery, Complication rate after surgery | December 2024 |
| NCT03034096 (GA-CARES) | Propofol vs. volatile anaesthesia | Cancer surgery | 1804 | All-cause mortality | Recurrence free survival, All-cause mortality as a binary outcome | December 2024 |
| NCT04800393 (TeMP) | Inhalation vs. Total Intravenous | Breast Cancer | 130 | NLR (1 h and 24 h) | Levels of multiple immune cells | April 2028 |

**Table 1.** *Cont.*

| Trial Number (Acronym) | Area of Focus | Type of Surgery | Number of Patients | Primary End-Point | Secondary End-Points | Est Year of Completion |
|---|---|---|---|---|---|---|
| NCT02840227 | Combined General/Regional vs. GA | Lung Cancer | 2000 | Cancer-free survival | Pain intensity, Opioid use | December 2023 |
| NCT04259398 | Propofol vs. Sevoflurane | Colon cancer | 792 | 5-year survival | Five-year RFS, One-year RFS | February 2026 |
| NCT03134430 | Regional Nerve Block on Cancer Recurrence | Gastric, colon, rectal, liver, lung | 400 | RFS | OS | May 2023 |
| NCT04601961 | Propofol vs. Sevoflurane | Colorectal cancer | 220 | HIF-1 gene expression | Number of recurrences, Gene expression (HIF-1, IL-6, TNF-alpha) | March 2024 |
| NCT04493905 (ENCORE) | Effects of anaesthetic techniques | Colorectal cancer | 10,000 | Time to Return to Intended Oncologic Therapy (RIOT) | Postoperative mortality for 0–30 days, Cancer recurrence 90 days, 1, 3- and 5-years | November 2023 |
| NCT04532606 | Remimazolam | Bladder Cancer | 1128 | Incidence of emergence delirium, recurrence free survival | Incidence of postoperative delirium, overall survival | October 2023 |
| NCT05742438 | Dexmedetomidine Infusion, Lidocaine Infusion, and Intrathecal Morphine Injection | Colorectal cancer | 114 | Plasma Matrix metalloproteinase-9 levels | Various immune cell levels | March 2024 |
| NCT05141877 | Propofol vs. sevoflurane | Primary brain tumour | 706 | Overall survival | Presence of disease progression | November 2025 |
| NCT04513808 | Propofol vs. sevoflurane | Oesophageal cancer | 950 | Recurrence free survival | The treatment effect of propofol-based anaesthesia versus volatile anaesthesia | December 2024 |
| NCT05250791 (FLICOR) | Lidocaine | Large bowel cancer | 50 | Feasibility of recruitment | Disease free survival | March 2024 |
| NCT04503148 | Propofol vs. inhalational anaesthesia | Renal Cell | 562 | 1-year metastasis-free survival | 3-year metastasis-free survival, 1-year survival | July 2025 |
| NCT04475705 | Propofol vs. sevoflurane | Paediatric tumour surgery | 100 | Difference in Hypoxia Inducible Factor-1 gene expression | Difference in levels of Interleukin-6/TNF alpha/CRP | July 2028 |
| NCT05484687 | Lidocaine | Colorectal cancer | 100 | Concentration of tumour micro metastasis markers | Concentration of stress hormones/inflammatory factors | December 2023 |
| NCT05663242 | Propofol vs. Sevoflurane | Primary lung cancer | 300 | Overall survival, presence of disease progression | Postoperative complications, hospital length of stay | November 2026 |

**Table 1.** *Cont.*

| Trial Number (Acronym) | Area of Focus | Type of Surgery | Number of Patients | Primary End-Point | Secondary End-Points | Est Year of Completion |
|---|---|---|---|---|---|---|
| NCT05331911 | Propofol vs. Sevoflurane | Primary Liver cancer | 500 | Overall survival, presence of disease progression | Postoperative complications, hospital length of stay | March 2027 |
| NCT05450055 | Postoperative Intraperitoneal Lidocaine | Ovarian cancer | 60 | Postoperative analgesic use | Survival time, Disease-free survival time | July 2029 |

## 8. Conclusions

While the research base examining onco-anaesthesia is expanding, it is still largely made up of laboratory investigations and observational retrospective studies which are inherently limited and cannot be used as the basis for a change in practice. There remains a relative paucity of sufficiently powered RCTs which are necessary to confirm a causal link between any perioperative intervention and long-term oncologic outcome. A summary of the main registered RCTs in this field is shown in Table 1. Results from ongoing RCTs, such as the CAN and VAPOR-C trial, are awaited. Future trials may require documentation of effects of anaesthesia–analgesia on precise subtypes of tumour, in addition to taking account of patient-specific tumour genomic analysis. In the absence of any convincing Level 1 that recommends a change in practice, long-term oncologic benefits should not be part of the decision when choosing an anaesthetic technique for tumour resection surgery. All currently available anaesthetic–analgesic techniques are valid for cancer patients, the choice of which should continue to be a shared decision between patient, anaesthesiologist and surgeon based on known risks and benefits.

**Author Contributions:** Writing—original draft preparation, L.M.; writing—review and editing, J.S. and D.J.B.; supervision, D.J.B. All authors have read and agreed to the published version of the manuscript.

**Funding:** This research received no external funding.

**Conflicts of Interest:** The authors declare no conflict of interest.

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
