# Peer review of "Anaesthetic Techniques and Strategies: Do They Influence Oncological Outcomes?"

_curroncol, doi:10.3390/curroncol30060403_

Round 1

Reviewer 1 Report

Dear authors, 

it has been a pleasure to read your review, that gets onto a topic that has become somehow fashionable. It is well-written and makes clear its point. Furthermore, Table I may prove of great interest for readers interested in the follow-up of the controversies here commented. As a non-clinician, I am grateful for the inclusion of 'wet-lab' studies. However, before recommending the study for publication, I would like to see some issues corrected. 

1. General comments: 

Type of study: This review is in many ways written as a systematic review, being however styled as a narrative one.  Although not necessarily explicited in the text, I would appreciate some kind of self-definition. If it has been conceived as a systematic review, a Table including keywords and databases used for each subject is mandatory, as well as a list of criteria for inclusion or exclusion. A narrative review would benefit from the removal of the text of some numerical data, to be placed in another Table. 

Quality of the studies and organization of the information: Although the text makes some attempts at evaluating the quality of evidence provided by each study, this is not performed in an homogeneous way, and there are some issues that remain un explained. Of particular interest for both oncologists and pathologists is the stratification of patients; in some studies regarding the issue patients are not classified regarding tumour sub-type and not even TNM and that issue should not be overlooked; also, the combination of drugs used in each clinical study is worth to mention (or to highlight if it has been unreported). A Table putting together the main features of each study (a single comprehensive one or a specific supplementary Table for each chapter) and evaluating any potential bias and the level of evidence provided would be a great improvement for the manuscript. 

2. Specific issues: 

Lines 114-116: this issue deserves to be justified more extensively

Lines 129-130: plese replace the adjectives by 'a large, multi-center'

Lines 144-147: ¿why should be a confounding factor the use of opioid in epidural infusion?, it seems that it may reinforce certain results, particularly if the authors intend to compare regional versus non-regional anaesthesia

Lines 155-169: It may be debatable, but it seems that a study where the primary endpoint has nothing to do with long term survival may not need such a thorough discussion in this paper. 

Lines 199-202: which histological type?

Lines 229-230: a short paragraph or a figure dealing more extensively with this complex signalling pathway would be appreciated. 

Lines 277-279: which was the primary endpoint?

Overall it is a comprehensive and informative study. 

Un saludo. 

Author Response

Thank you for your meticulous review. We have implemented all the specific tasks you mentioned in your list.

Regarding the type of review, we clarify that this is indeed a Narrative Review, never intended as a systematic review. We have included actual summary data from most of the important studies, believing that this is essential to give readers an indication of the type of data and the size of the effect measured. We therefore request that these details are retained in the body of the manuscript itself, consistent with normal practice among journals for clinical reviews.

We appreciate your comments on the quality of the studies, whether tumour subtypes were formally stratified, and a new table displaying all the features of every study. Where studies highlighted tumour subtypes, we have mentioned this, but in truth few of the available studies have cited do so. One of the points we have made is that future studies should in fact, pay more attention to the effect of anaesthetic technique on various tumour subtypes and on individuals’ tumour genetic expression. We think that a table listing each of our references and grading the evidence of each in a separate column would be unwieldy and challenging to render accessible for readers.

Specific issues highlighted:

Lines 144- 147: We are attempting to highlight that comparing regional v’s non-regional anaesthesia can be confounded by the effects of opioids if used as part of a regional anaesthesia technique.

Lines 155-169: While it would be favourable to only use studies whose primary end-points were directly related to long-term survival or oncological outcomes, we believe this study does provide useful information on its long-term analysis of its secondary end-points. We have highlighted the limitations of this study in the text.

Lines 277-279: Neither of the end-points discussed in the text were the primary end-point. The primary end-point in this study was “natural killer (NK) cell activity, which was measured preoperatively and 1, 3, and 5 days postoperatively.”

Reviewer 2 Report

Dear authors,

thank you for the opportunity of read and review this manuscript. The topic is interesting.

The manuscript is clear and well written, I have just few recommendations to improve its quality.

A short abstract should be added to better present the paper.

Figure 1 is very interesting; line 33-35 should be discussed in the text and not in the figure legend.

Method section should be added to explain how research was performed.

Reference 8 in opioids section (line 56) should be checked as it mostly focuses on k opioid receptor expression in breast cancer (in vitro) instead of mu.

Author Response

Thank you for your meticulous review. We have altered the text where indicated in your comments. We have checked the 8th reference in the opioid section and edited the text appropriately.

Reviewer 3 Report

This review is well-written, and I have just a few suggestions or comments.

Authors should describe the effects of opioids for immune cells in terms of cancer immunology.  

Does the propofol and local anesthesia have direct effects to cancer cells?

Authors should describe the mechanism of action how COX-2 inhibitor affects cancer cells and surrounding immune cells in more details.  

Author Response

Thank you for your meticulous review.

Authors should describe the effects of opioids for immune cells in terms of cancer immunology

We have added a few sentences to elaborate on this but not in too much detail. 

Does propofol and local anesthesia have direct effects to cancer cells?

We have outlined where amide local anaesthetics in particular effect cancer cell biology.

Authors should describe the mechanism of action how COX-2 inhibitor affects cancer cells and surrounding immune cells in more details.

Again we have added some details about this but believing the readership to be largely clinicians, we have not elaborated to a great extent.